# Long-Term Influence of Locus of Control and Quality of Life on Metabolic Profile in Elderly Subjects with Type 2 Diabetes

**DOI:** 10.3390/ijerph192013381

**Published:** 2022-10-17

**Authors:** Annalisa Giandalia, Marta Ragonese, Eugenio Alessi, Maria C. Ruffo, Alberto Sardella, Alessandro Cuttone, Maria A. Aragona, Antonio G. Versace, Giorgio Basile, Domenico Cucinotta, Giovanni Squadrito, Giuseppina T. Russo

**Affiliations:** 1Department of Clinical and Experimental Medicine, University of Messina, 98125 Messina, Italy; 2Department of Human Pathology DETEV, University of Messina, 98125 Messina, Italy; 3Grande Ospedale Metropolitano “Bianchi, Melacrino, Morelli”, 89124 Reggio Calabria, Italy; 4Presidio Ospedaliero “G. Jazzolino”, 89900 Vibo Valentia, Italy

**Keywords:** type 2 diabetes, Locus of Control, elderly, quality of life, gender differences

## Abstract

Background: The Locus of Control (LOC) is a mental disposition indicating the individuals’ belief that disease-related outcomes are under their own control (Internal), dependent on others (External), or dependent on chance (Chance). Quality of Life (QoL) and LOC may have complex effects on self-care activities and diabetes management in subjects with type 2 diabetes (T2D). The aim of the present study was to evaluate the predictive role of LOC and QoL scores on metabolic control in elderly T2D outpatients, secondly evaluating potential gender differences. Methods: An extensive set of questionnaires was administered to a group of consecutive elderly T2D outpatients on oral glucose-lowering drugs attending a single diabetes center. Personal and clinical variables were analyzed at baseline (between 1 February and 31 March 2015) and after 6 years of follow-up. Results: At baseline, study participants showed an overall good metabolic control. Diabetes Specific Quality of Life (DSQoL) scores indicated an overall good QoL in both genders, with a higher DSQoL satisfaction score in women. Both genders presented higher scores in the LOC-Internal domain, with men reaching higher scores in the LOC-External domain than women. At the 6-years follow-up, subjects with baseline higher LOC-External score presented better metabolic outcome. In the regression analysis, LOC-External score was an independent predictor of good metabolic control maintenance, but this result was only statistically significant in men. Conclusions: LOC scores may influence long-term glycemic control in elderly T2D patients on oral glucose-lowering drugs.

## 1. Introduction

The rising prevalence of type 2 diabetes (T2D) represents a public health challenge worldwide, especially in older adults. T2D profoundly affects various aspects of daily life in elderly subjects, including quality of life and psychological well-being. Due to the intrinsic age-related characteristics, elderly subjects with T2D require a personalized treatment, which takes into account their comorbidities, subjective risk of hypoglycemia, cognitive and the other domains of health status [1,2].

Therefore, in addition to physical health, a people-centered management of chronic patients, in particular of elderly diabetic subjects, must also include the evaluation of adherence to therapy, independence in activities of daily living (ADL) and psychological status.

Current T2D guidelines for the elderly also recommend the preservation of Quality of life (QoL) overtime among the goals of diabetes treatment, in addition to the prevention of long-term complications [1].

Quality of Life (QoL) is a subjective concept, which can be measured by reliable instruments with a good analytical power [3]. Notably, QoL may be strongly influenced by T2D due to the need for frequent clinical check-ups, dietary and lifestyle changes, and numerous data suggest that individuals with T2D have poorer QoL, potentially contributing to inadequate care management [4,5]. On the other hand, QoL may itself influence diabetes management, in particular by affecting patients’ adherence and compliance.

While the role of QoL in diabetes management has been increasingly recognized, other psychological factors may influence patients’ behavior and their adherence to diabetes care, including the Locus of Control (LOC), an expression that literally means “place through which control is exercised”. LOC is a mental disposition, and includes 3 domains: Internal, External (or Powerful Others), and Chance [6]. These domains indicate and measure the patient’s belief that diabetes and diabetes-related outcomes (i.e., hypoglycemic episodes and glucose control) are under their own control (Internal), dependent on others (External), or dependent on chance/fate (Chance) [7,8].

LOC domains may affect a patient’s mood and psychological well-being; furthermore, they may impact on the approach to diabetes management, in its multifaceted aspects (daily self-management, dietary changes, medications), potentially influencing clinical and psychological outcomes [6,7,8]. For example, the tendency to delegate control or to concentrate all responsibilities on oneself may direct choices and behaviors related to T2D management.

However, to date, the impact of LOC domains on diabetes management and outcomes has not yet been fully investigated, especially in elderly patients, who are more vulnerable to the risk of anxiety and mood disorders.

Moreover, the impact of QoL and LOC in T2D management and outcomes may be affected by gender. Thus, men and women behave differently in relation to various aspects of the disease, including psychological aspects [9]. Several sex- and gender-differences have been described in the field of T2D, including clinical outcomes and chronic management of the disease [10,11]. Notably, the numerous differences observed between T2D men and women on metabolic control, achievement of glycemic and extra-glycemic target, and the risk of chronic complications, particularly cardiovascular disease (CVD), are attributable to the complex interaction of biological, genetic and hormonal differences, with non-biological factors, including psychological, social and family variables [10,11]. In particular, metabolic control is overall unsatisfactory, since only half of T2D patients reach recommended HbA1c values, but T2D women experienced more difficulties in reaching them; furthermore, blood pressure control is more often impaired in T2D men whereas severe obesity is more frequent in T2D women, who also show a higher relative risk of developing cardiovascular complications [10,11].

Notably, gender-differences have also been observed in cognitive status and LOC, showing a greater impact of diabetes-related distress on metabolic control in T2D women than in T2D men [9].

However, to date, the impact of QoL and LOC on T2D management has not yet been fully clarified, especially in elderly T2D subjects. In particular, whether elderly patients’ control orientation (LOC domains) and QoL might influence long-term metabolic, and micro- and macro-vascular diabetes outcomes, their potential relationship with QoL, and whether these associations are different in elderly men and women, deserves further investigation.

Therefore, the purpose of this study was to determine the long-term predictive role of QoL and LOC on metabolic control and T2D long-term complications in elderly T2D outpatients observed for 6 years, secondly taking into account any potential gender differences.

## 2. Materials and Methods

### 2.1. Study Population

All consecutive T2D subjects on oral hypoglycemic agents aged ≥65 years attending the Metabolic Disease Outpatient Clinic of the University Hospital of Messina, (Sicily), Italy, between 1 February and 31 March 2015, were included in this single-center observational study.

The following exclusion criteria were applied: T2D diagnosis for less than two years, treatment with hypoglycemic injectable therapy (either insulin or GLP-1 RAS) or with diet only, and having co-morbidities such as active cancers, chronic renal failure in the hemodialysis phase, severe sensory deficits, severe psychiatric or neurological conditions. Subjects with severe cognitive impairment (as defined by a Mini Mental State Examination score < 18) were also excluded.

All study subjects were of Caucasian ethnicity. They regularly attended the outpatient diabetes clinic of the University Hospital of Messina (Sicily) in Italy. The clinic is part of a university hospital, that is located in in an urban area, and all patients regularly attend the clinic at least twice a year.

### 2.2. Clinical and Laboratory Parameters

Information on the level of school education, duration and complexity of T2D management was also collected.

At baseline, information on the level of school education was expressed in years and the following classification was applied according to the Italian educational system: 1–5 years, elementary level; 6–13 years, high-school level; >13 years, University degree.

Duration of the disease was collected since T2D diagnosis, and the complexity of diabetes management was defined on the basis of the number of specialist outpatient visits in the previous two years. The presence of hypoglycemia was defined on the basis of at least one episode of hypoglycemia in the six months preceding the study.

Weight, BMI, waist circumference, glycemic control, lipid profile and creatinine levels, as well as any ongoing chronic therapy and the presence of diabetes chronic complications, were analyzed at baseline and after 6 years of observation.

### 2.3. Psychometric Evaluation

At baseline, all study subjects underwent an extensive psychometric evaluation: cognitive status, functional autonomy, patients’ satisfaction with their diabetes treatment, Quality of Life (QoL) and Locus of Control (LOC) were explored by validated questionnaires. All study subjects responded to self-administered questionnaires by handwriting their responses.

The cognitive status was explored in all patients at baseline, by the Mini Mental State Examination (MMSE).

The MMSE is a brief screening test for the assessment of global cognitive status [12]. It assesses attention, orientation, language, immediate and short-term recall, and the ability to perform simple written and verbal commands. The score ranges from 0 to 30 points. A score ≤ 18 suggests a severe impairment of cognitive abilities; a score 18–24 indicates moderate to mild impairment, a score of 25–30 falls within the normal ranges. Subjects with a MMSE score < 18 were excluded from the analysis.

The functional status was investigated by the basic Activities of Daily Living (ADL) and Instrumental Activities of Daily Living (IADL) scales [13,14]. We expressed the values of ADL and IADL as number of maintained functions: low ADL and IADL scores indicate difficulties in physical and/or cognitive health.

The Katz Activities of Daily Living (ADL) Assessment Form [13] explores the capability of older adults to autonomously perform 6 basic self-care tasks that include walking, feeding, dressing, bathing, transferring. The score ranges from 0 (minimal autonomy) to 6 (full autonomy).

The IADL scale evaluates the performances in 8 instrumental daily activities in older people, including using the telephone, shopping, meal preparation, cleaning house, managing finances, communication, transportation, and medications [14].

The score ranges from 0 (minimal Instrumental Activities autonomy) to 8 (full Instrumental Activities autonomy).

### 2.4. Diabetes Specific Quality of Life

The Diabetes Specific Quality of Life (DSQoL) questionnaire was originally used in the late 1980s in the Diabetes Control and Complications Trial (DCCT), to assess the impact of diabetes on QoL; the Italian validated version of the questionnaire [15] was administered at baseline to each patient.

The DSQoL evaluates the impact of diabetes on a patient’s life, the degree of satisfaction with the treatment and the extent of concern about chronic complications.

The questionnaire consists of 39 questions divided into three evaluation domains: satisfaction (14 questions, the score ranging from 14 to 70), impact (20 questions, the score ranging from 20 to 100), and worry (5 questions, the score ranging from 5 to 25). Lower scores in the three examined domains are an expression of a better quality of life.

### 2.5. Locus of Control

The Locus of Control (LOC) questionnaire is used to determine the attribution of responsibility by the patient for the evolution of diabetes [7].

The questionnaire consists of 18 items investigating three different evaluation domains, with scores ranging from 6 to 36: the Internal domain, in which the responsibility is attributed to internal factors (such as self-efficacy and commitment), the External domain, in which the responsibility is attributed to the involvement of external factors or the involvement of other people (e.g., doctors, nurses and family members), and the Chance domain, in which the responsibility is attributed to destiny, luck and chance.

The highest test score is the expression of the predominant domain of control.

### 2.6. Diabetes Treatment Satisfaction Questionnaire

The Diabetes Treatment Satisfaction Questionnaire (DTSQ) evaluates a patient’s satisfaction to with their diabetes treatment [16], regardless of the type of treatment (diet, therapy with oral hypoglycemic agents or insulin therapy).

The questionnaire consists of 8 items, the scoring based on a Likert scale from 0 (e.g., “very dissatisfied”, “very inconvenient”) to 6 (e.g., “very satisfied”, “very convenient”) for each question. Six questions measure treatment satisfaction, and ask about “satisfaction with current treatment”, “flexibility”, “convenience”, “understanding of diabetes”, “recommend treatment to others” and “willingness to continue the current treatment”. These six scores are added up to produce a DTSQ total score (range 0–36): higher scores indicate higher treatment satisfaction (total DTSQ score 36= very satisfied).

DTSQ items 2 and 3 are rated differently and evaluate the burden from hyper- and hypo-glycemia (0 being “none of the time” to 6 being “most of the time”).

DTSQ is broadly used, and officially approved by WHO and the International Diabetes Federation [16,17].

### 2.7. Assessment of T2D Chronic Complications

Both micro- and macro-vascular complications of T2D were screened according to national and international diabetes guidelines [18,19].

Macrovascular disease: coronary heart disease was defined on the basis of clinical documentation and of the reports of cardiologist specialists and/or hospital discharge (myocardial infarction, chronic ischemic heart disease, coronary heart by-pass, coronary angioplasty); a standard electrocardiogram and a cardiologist visit are performed annually in all T2D patients as part of the usual screening program. Cerebrovascular disease and peripheral arterial disease were assessed by color-doppler ultrasonography by B-mode real-time ultrasound, as part of the periodic screening of macrovascular complications.

Microvascular disease: diabetic retinopathy was diagnosed based on direct ophthalmoscopy performed by an expert ophthalmologist and/or by fluorescein angiography within 1 year before the start of study.

Diabetic kidney disease was assessed according to albuminuria measurement and estimation of Glomerular Filtration Rate (eGFR) by CKD EPI formula [20].

### 2.8. Statistical Analysis

Statistical analysis was performed using IBM SPSS (Statistical Package for the Social Science) version 26 (Armonk, NY, USA). Continuous variables were expressed as means ± standard deviation (SD). Categorical variables were expressed as number of cases and percentages. Since the majority of the investigated variables were normally distributed, as verified by the Kolmogorov-Smirnov test, a parametric approach was applied; the chi-square (χ²) test for categorical measures was used for comparisons. The analysis of variance (ANOVA) test was used to compare continuous variables. A hierarchical adjusted logistic regression was conducted to test the contribution of several variables to the maintenance of good metabolic control; precisely, socio-demographic and clinical variables were included in the first step, the psychological variables in the second step.

A post-hoc power analysis was performed, by using the G*Power software (version 3.1.9.6; Franz Faul, Edgar Erdfelder, Axel Buchner, Albert-Georg Lang, Germany); with a determined effect size of 0.30, a power (1-β error prob) of 0.86 was calculated.

## 3. Results

### 3.1. Personal and Clinical Characteristics of the Study Participants at Baseline, according to Gender

A total of 108 elderly T2D subjects agreed to participate in the study and answered the set of questionnaires at baseline. Among them, four patients were lost at follow-up (one patient died, 3 patients no information was no longer obtainable); therefore, the present analysis covers the 104 subjects with complete clinical information available at the time of follow-up.

As shown in Table 1, at baseline, study participants (61% men, 39% women) had a mean age of 72 years and a mean level of school education of 9.27 years, without significant differences between the two genders.

The mean duration of diabetes was of 12.5 years; study participants were overweight (mean BMI 27.21 ± 6.42 kg/m^2^, mean waist circumference 100.80 ± 11.39 cm), and presented good metabolic control (mean HbA1c 6.7%), with 71% of them showing HbA1c levels ≤ 7%. Men had higher mean body weight compared to women (76.18 ± 15.69 vs. 68.18 ± 13.15 kg, *p* = 0.008); conversely, BMI, waist circumference values and HbA1c levels were similar in the two genders.

Mean transaminase levels were within the normal range, and men had higher levels of GGT (34.38 ± 23.49 vs. 23.45 ± 10.68 U/L, *p* = 0.040). As for lipid profile, mean levels of HDL-cholesterol (49.32 ± 12.85 mg/dL) and triglycerides (130.02 ± 60.45 mg/dL) were similar in men and women; total cholesterol levels were higher in women (170.53 ± 29.97 mg/dL) than in men (159.11 ± 28.05 mg/dL), although this difference was not significant (*p* = 0.058).

Men had higher levels of creatinine (*p* = 0.007), although eGFR did not differ in men and women (67.38 ± 26.78 vs. 70.05 ± 23.93 mL/min/1.73 m^2^, *p* = 0.606).

At baseline, 68% of the study subjects (n = 71) had chronic complications, without any differences between men and women: 18 patients (17.3%) were affected by micro-vascular complications, 43 subjects (41.3%) had only macro-vascular complications, and 10 patients (9.6%) had both micro- and macro-vascular complications.

The mean number of specialist outpatient visits in the previous 2 years, a measure of the intensity of the care, and the percentage of subjects reporting at least 1 episode of hypoglycemia in the six months preceding the study were also similar in the two genders. As for T2D treatment, the majority (87%) of patients were on metformin, 12.5% on acarbose, 29.8% on secretagogues, 23.1% on DPP-4 inhibitors (DPP-4i) and 2.9% on pioglitazone, without any gender differences.

### 3.2. Gender Differences in Cognitive Functioning, Functional Status, Quality of Life and Locus of Control

Table 2 shows the baseline scores of the MMSE, ADL, IADL, DTSQ, dsQoL and LOC tests, according to gender.

Women showed lower MMSE scores compared to men (24.83 vs. 26.11, *p* = 0.028). Women also showed lower ADL scale scores as compared to men (5.58 ± 0.76 vs. 5.95 ± 0.43, *p* = 0.002); conversely, IADL scale scores were similar in the two genders (7.53 ± 1.24 vs. 7.75 ± 0.57, *p* = 0.223).

The DTSQ score indicated a good level of treatment satisfaction and a low burden of hyper- and hypo-glycemia, without differences between the sexes.

Mean DSQoL scores indicated an overall good quality of life in both genders, with a higher DSQoL satisfaction score in women as compared to men (*p* = 0.012), suggesting a better quality of life in men.

As for LOC, there was a high prevalence of the Internal domain over the other two domains in both genders (29.49 ± 6.92 in women, 29.56 ± 5.97 in men, *p* = 0.958), indicating a good awareness of the patient’s role in diabetes management.

Men presented higher scores in the LOC-External domain as compared to women (25.49 vs. 23.32, *p* = 0.048), suggesting a greater perception of the impact of external factors.

Scores in the Chance domain were the lowest LOC score in both genders (18.83 ± 7.84 in women, 17.87 ± 7.65 in men, *p* = 0.539).

### 3.3. Variation of Clinical Characteristics of the Study Participants after 6 Years of Follow-Up

Table 3 shows clinical characteristics of the study participants after 6 years of follow-up. Overall, mean BMI and body weight values decreased (*p* < 0.05) while HbA1c levels did not show any significant variation overtime; as for renal function, creatinine levels increased (+0.16 mg/dL, *p* < 0.001) and eGFR values showed a mean 5 mL/min reduction (*p* > 0.05). At follow-up, mean HbA1c was 6.66%, with 59% of study subjects showing at-target values (HbA1c ≤ 7%). After 6 years, most of the subjects maintained a good glycemic control, whereas 30 subjects (28.8%) required an intensification of hypoglycemic therapy (13 women, 31.7% and 17 men, 27.0% *p* > 0.05), defined as the increase in the number of prescribed OHAs and/or the prescription of injection drugs (insulin or GLP-1 RAs).

The first occurrence of chronic diabetes complications (micro- and/or macro-vascular complications) was recorded in 27 subjects (26%); in particular, 4 subjects were diagnosed with retinopathy, 5 patients with diabetes kidney disease (DKD), 16 with CVD and 2 subjects with both CVD and DKD.

When we stratified the study population according to gender, we observed the same trend in the variation of study variables at follow-up in men and women, without any between-gender difference; this was also the case in the occurrence of chronic complications (26.8% vs. 28.6% *p* > 0.05) (Appendix A).

Notably, no significant differences in baseline DSQoL and LOC scores were noted between subjects developing or not developing new chronic (micro- and/or macro-vascular) complications (Appendix A).

### 3.4. Factors Associated with Maintenance of Good Glucose Control Overtime: Role of Quality of Life and Locus of Control

At the end of the follow-up, all of the study participants were divided into two groups, according to the course of their glucose control over time. In particular, those who showed a worsening of glycemic control (HbA1c values ≥ 7%) overtime and/or required an intensification of hypoglycemic treatment during the 6-years observations, were grouped in the <worsening control> group (n = 64); study participants improving their glycemic control or maintaining a good glycemic control over time (HbA1c ≤ 7%), without any intensification of hypoglycemic treatment, were grouped in the <maintaining good control> group (n = 40).

Baseline clinical characteristics and scores of the Mini Mental State Examination, ADL, IADL, DTSQ, DSQoL and LOC tests of these two groups are shown in Table 4.

No significant differences emerged between the two groups in age, duration of diabetes, mean values of BMI and waist circumference; mean creatinine levels and eGFR values were also similar in the two groups. The <worsening control group> had higher mean HbA1c levels at baseline than the <maintaining good control> group (6.92 ± 0.61 vs. 6.39 ± 0.60, *p* < 0.001).

Regarding the questionnaires, the two groups had similar scores in the MMSE, IADL, DTSQ and QoL. Subjects in the <maintaining good control group> had lower ADL score 5.60 ± 0.87 vs. 5.98 ± 0.38, *p* = 0.0339) compared to subjects in the <worsening control group>.

Significant differences were also noted in baseline LOC scores domains in the two groups (Figure 1). In particular, subjects in the <maintaining good control> group had significantly higher mean LOC-External score compared to subjects in the <worsening control> group (26.40 ± 4.47 vs. 23.11 ± 4.79 *p* = 0.0007), while the LOC-Internal and LOC-Chance scores were similar in the 2 groups.

Similar results were observed when this analysis was performed for in men and women, separately: women maintaining a good glycemic control reported a significantly higher baseline LOC-E score (25.88 ± 6.74), compared to those with a worse glycemic control (mean 21.68 ± 5.32, *p* = 0.032). Accordingly, men maintaining a good glycemic control reported significantly higher LOC-E score (27.31 ± 4.43), compared to those in the <worsening control> group (24.22 ± 5.52 *p* = 0.021). Other LOC domains were not significantly different in the two subgroups.

A hierarchical adjusted logistic regression was performed to assess the associations between the maintenance of good metabolic control overtime and all study variables.

At multivariate regression analysis, baseline-HbA1c levels (B −2.358, *p* < 0.001, ExpB 10.569, 95%CI 3.138–35.596), ADL (B −1.70, *p* = 0.017, ExpB 0.183, 95%CI 0.045–0.738) and LOC-E scores (B 0.255, *p* = 0.001, ExpB 1.291, 95%CI 1.112–1.498) were independent predictors of good metabolic control maintenance in the study population (Appendix A).

When we stratified by gender the study population, baseline-HbA1c levels (B 2.265, SE 0.737, *p* = 0.002, ExpB 9.628 95%CI 2.270–40.837) and LOC-E score (B 0.179, SE 0.081, *p* = 0.027, ExpB 1.196 95%CI 1.020–1.403) were independent predictors of good metabolic control maintenance in men; none of the analyzed variables were associated with good metabolic control maintenance in women.

## 4. Discussion

Diabetes management should not only assess the achievement of treatment measures, but it should also include the evaluation of some psychological aspects, including wellbeing, treatment satisfaction, QoL and, possibly, LOC. This is especially true in elderly subjects.

In our study we assessed the effects of specific QoL and LOC domains on metabolic control and long-term diabetes complication in a group of elderly subjects, attending a single diabetes centre in Southern Italy who were followed for 6 years.

At the basal evaluation of our study population, we observed higher scores in the Internal and External LOC domains, over the Chance domain, which showed the lowest score; this suggests that our patients have a tendency to attribute the cause of diabetes-related events to themselves and to their caregivers, rather than to fate or chance. Notably, our study also demonstrated that the LOC-External domain had an independent impact on long-term glycemic control, even after multivariate adjustment, together with baseline HbA1c levels and ADL scores; conversely, QoL scores were overall low, but they did not independently influence long-term metabolic control in our study population. Thus, elderly patients with higher LOC-External scores at baseline were more likely to improve or maintain a good glycemic control over the six years of observation. Subjects with a higher LOC-External score tend to attribute the “control” of the disease to other people (doctors, nurses, therapists, psychologists, family members, caregivers), demonstrating a greater sense of trust and willingness to accept outside help. Current guidelines on T2D management recommend the active involvement of the patient in the treatment plan, with the aim of making them aware and active. Although our study is not conclusive, our results suggest that a predominantly LOC-External could have a positive value in specific subgroups of patients, and we may speculate that, in elderly subjects, this attitude may lead to a lower level of stress and to a greater adherence to the treatment plan, with less impact of anxiety and depression. Consistently, in our elderly population, higher LOC-External scores were associated with improved glucose control overtime, as assessed by HbA1c values.

Conversely, we may hypothesize that higher LOC-Internal domain scores might be more advantageous in younger adults, indicating self-independence in T2D management, whereas relying on oneself for T2D-related activities may cause anxiety or concern in elderly subjects.

Importantly, LOC scores should not be interpreted in a rigid or inflexible way, since each LOC domain can have a positive value, and none is, in absolute terms, more functional or adaptive over the other two [21,22,23]. The external LOC may therefore hesitate in an attitude of less active involvement towards chronic disease, but also of greater trust and acceptance of help by health- and rehabilitation-personnel and family members [22]. Moreover, an external LOC may help individuals to minimize their responsibility in the occurrence of negative events, giving an external explanation of what happens and reducing the sense of guilt [22,23].

Some of the previous studies evaluated the impact of LOC on glucose control in adult subjects with diabetes, with conflicting results. Thus, some authors found a positive relationship between Internal LOC and diabetes management [24], conversely, others found a positive relationship between External or Chance LOC and better glycemic control [6,25,26]; finally, other studies did not find any significant relationship between the 2 variables.

Moreover, in 2010, a meta-analysis of 17 studies (published between 1985 and 2006) investigated the effect of LOC on diabetes control in a very heterogenous population of adult patients with T2D or T1D, suggesting that there is only a slightly positive correlation between External and Chance LOC domains with metabolic control of diabetes [27].

However, these studies differed from each other by in their sample sizes, assessment of clinical characteristics and measures of LOC. These discrepancies may also depend on the heterogeneity of the investigated study populations, in various aspects, including ethnicity, diabetes treatment and age.

However, the available data do not seem to suggest an impact of age, ethnicity or geographic origin on the relationship between LOC and diabetes control; other clinical and social variables could therefore play a more decisive role.

Moreover, at variance with our study, the majority of these studies were cross-sectional, and the few longitudinal ones had a very short follow-up observation, and none of them specifically addressed elderly T2D subjects, or the evaluation of evaluating potential gender-specific differences, as in our analysis.

Moreover, the time frame of the investigation should be taken into account, since the the majority of the reported studies are not recent and diabetes management has profoundly changed over the last decades, thanks to the availability of more efficacious hypoglycemic drugs with low hypoglycemic risk.

Our study also demonstrated that metabolic control in elderly T2D subjects is independently influenced by their capability to perform basic self-care tasks, expressed by ADL scores. Our study subjects had high baseline scores of IADL, ADL, and MMSE, indicating a high level of autonomy and cognitive function.

Notably, subjects included in our study were all people living in an urban context, far from a rural setting, with easy access to hospital facilities (they attended the clinic at least twice a year) and with a relatively high level of education (>9 years), without significant differences between the two genders; these aspects may all potentially influence QoL and LOC domains and should be taken into account when interpreting our data.

The inclusion of exclusively elderly subjects is another relevant point, since age may modulate the impact of LOC and QoL on glucose control. Diabetes has a profound impact on patients’ psychological health and some studies found an increased risk of psychosocial distress and depression among younger adult T2D patients, than in their older counterparts. The greater psychological impact of diabetes in younger adults could be also due to social factors and it suggests a greater need for of psychosocial and self-management support in younger T2D patients than in older subjects [28,29,30]. As for the impact of LOC in the different ages of life, we may hypothesize that in young people, a predominant Internal LOC domain could be more useful, resulting in greater awareness, active involvement and commitment in the care plan; on the contrary, in the elderly, an attitude of greater trust and therefore greater adherence (external LOC domain) could be advantageous.

Another aspect to consider is that our study is mono-centric, therefore our patients were habitually followed at the same second level diabetes outpatient clinic, undergoing at least two specialist visits throughout the year, and this aspect might have influenced the perceived impact of external help on diabetes management. Thus, the doctor-patient relationship is an acknowledged factor influencing adherence and consequently glucose control [31]. In line with this hypothesis, elderly subjects in our study were overall satisfied with the plan of care, as shown by the DTSQ (Diabetes Treatment Satisfaction Questionnaire) scores [16]; they also showed a low-medium burden from hyper- and hypo-glycemia, as indicated by mean scores in DTSQ 1 and 2 items; a good level of satisfaction may positively influence adherence and metabolic control, also with a lower impact on QoL.

In addition, the type of treatment and the achievement of glucose targets are relevant issues. Study subjects were overall in fair/good glycemic control and reported few episodes of hypoglycemia, all characteristics that can profoundly influence QoL and LOC. According to the study design, we selected elderly subjects on OHA in order to exclude the potential bias of injectable therapies, including insulin therapy and the consequential influence of intensive glucose monitoring and hypoglycemic risk on QoL and LOC. Moreover, the majority of them were on metformin and only few required an intensification with insulin after follow-up. Certainly, diabetes treatment as well as adherence to dietary advice have an important role in glucose control overtime, but in our relatively stable T2D elderly patients, hypoglycemic drugs were not independent predictors of HbA1c variation at follow-up.

Thus, the majority of our patients were already on a good control at baseline and were able to maintain a good control during the follow-up. This aspect is very important because maintaining at target HbA1c levels overtime is essential to prevent long-term complications [18].

Our study also demonstrates that QoL and LOC did not influence the risk of developing chronic complications. In this regard, it is likely that the overall low incidence of new chronic complications, the monocentric nature of our study, and the relatively small sample size played a role.

Notably, the multivariate analysis, testing all the study variables, showed that, together with higher LOC-External scores, baseline-HbA1c levels and lower ADL scores were independent predictors of good metabolic control maintenance in our population.

The role of baseline HbA1c in the achievement of long-term glucose control is well-documented and it has been recently confirmed as the main determinants of the achievement of HbA1c levels < 7 and no weight gain also with the machine learning technique in the data of over 1.5 million patients [32].

As for ADL scores, metabolic control in elderly subjects could be influenced by their capability to perform basic self-care tasks and, in our population, differences in ADL were noted between the “worsening” and the “maintaining glucose control” groups.

It is important to point out that study subjects had mean ADL and IADL scores indicative of preserved functional abilities. Elderly people with diabetes are at increased risk of frailty and disability [33,34], which in turns are associated to with adverse health outcomes, including mortality risk in older subjects [35]. In addition, a lower functional ability can translate into a greater tendency to rely on caregivers and therefore it may influence LOC domains.

Finally, our study explored potential gender differences in the impact of QoL and LOC on metabolic control and long-term diabetes complications in this elderly population, because of the numerous differences in clinical manifestations, risk factors, outcomes and psycho-social aspects reported between T2D men and women [9,10,11,36]. The MMSE scores, as well as the ADL and IADL scales, suggested lower functional and cognitive performances in T2D elderly women than in men, although within a range suggestive of healthy status. Our results are in line with those of other studies that have showed a greater likelihood of women with diabetes to have cognitive impairment and poor physical function outcomes [37,38].

Notably, the IADL evaluation might be affected by gender, due to social and culture-related aspects; however, no significant differences were noted in our study in baseline IADL scores between men and women (Table 2). Moreover, our sample showed a generally preserved autonomy in performing instrumental daily activities; thus, we might be confident that, at least in our sample, the influence of gender could be not relevant.

We also observed a better QoL and higher external LOC domain scores in T2D men, indicating that the two genders may suffer differently due to the impact of diabetes and may respond differently in terms of self -management. In line with recent reports, our results suggest that not only genetic and metabolic factors, but also social and psychological variables determine the known differences in terms of goal achievement in T2D men and women; functional limitations and behavioral factors, could make an indirect contribution to gender disparities in clinical outcomes [9].

We cannot explore the contribution of sex-related variables including the role of estrogens, since our results refer to a population of elderly T2D subjects and we cannot extend them to younger females with preserved hormonal milieu. On the other hand, in elderly subjects it is likely that “gender” differences in social, behavioral aspects would prevail in determining the observed differences in QoL and LOC scores.

Several limitations must be taken into account when evaluating our results. In particular, we acknowledge that the relatively small sample size might narrow the generalizability of our findings, despite the performed post-hoc power analysis. Another limitation is the lack of LOC evaluation at follow-up. However, LOC could be generally considered a dispositional trait, therefore no significant changes are expected over time and a single assessment may provide a reliable estimate of the LOC of adults [39,40]. Moreover, our study population consists of autonomous subjects and this condition may limit the extensibility of our results to more heterogeneous and different populations.

Another limitation of the study is the absence of information on several important variables such as marital status, income, family status (living alone/co-habitat/care facility), which could influence their LOC and QoL, and the lack of information on behavioral changes, since change in diet and physical activity overtime may affect metabolic control; however, on this aspect, subjects participating to in our study were all elderly people with a mean diabetes duration of 12 years, all regularly followed at our center and we may hypothesize that diet and physical activity did not significantly change over time in this population of elderly patients with a chronic disease.

The extensive psychometric evaluation, the accurate evaluation of chronic micro- and macro-vascular complications, the length of follow-up, and the periodic regular assessment of our outpatients are among the strengths of the study.

## 5. Conclusions

In conclusion, these results suggest that in elderly patients on oral glucose-lowering drugs, attending a single third-level facility of diabetes care in Southern Italy, with a good QoL, LOC scores may influence long-term metabolic control.

If confirmed on a large scale, our results may have important clinical implications. They suggest that, in outpatient clinical practice, the evaluation of important psychological and psycho-social variables in subjects with T2D, especially if elderly, could be useful to create a personalized approach focused on the subjective needs of patients. In their everyday life, patients constantly reformulate the prescribed therapy based on their personal concept of disease, on their perception of the cure, and on their Locus of Control. Knowing the patient’s LOC can allow the physician to formulate shared and actionable requests.

## Figures and Tables

**Figure 1 ijerph-19-13381-f001:**
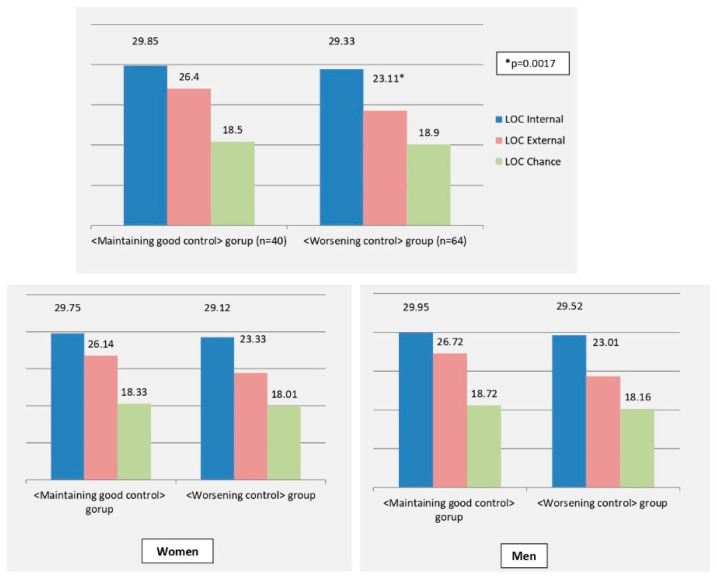
LOC scores in the <Worsening control> and <Maintaining good control> groups. <**Worsening control**> group (n = 64): subjects who over time presented a worsening of glycemic control and/or requested an intensification of hypoglycemic therapy vs. baseline. <**Maintaining good control**> group (n = 40): subjects who over time improved their glycemic control or maintained a good glycemic control.

**Table 1 ijerph-19-13381-t001:** Clinical characteristics at baseline of T2D elderly subjects participating in the study, according to gender.

Clinical Characteristics at Baseline	All	Women	Men	*p*
n	104	41 (39.4)	63 (60.6)	
Age (years)	71.88 ± 6.95	70.56 ± 5.91	72.75 ± 7.46	0.117
Level of school education (years)	9.27 ± 3.75	9.00 ± 3.87	9.44 ± 3.69	0.557
Diabetes duration (years)	12.50 ± 9.74	12.54 ± 11.29	12.45 ± 8.78	0.817
Weight (kg)	73.73 ± 13.43	68.18 ± 13.15	76.18 ± 15.69	**0.008**
BMI (kg/m^2^)	27.21 ± 6.42	27.89 ± 5.26	26.77 ± 7.07	0.388
Waist circumference (cm)	100.80± 11.39	98.56 ± 10.78	103.04 ± 11.76	0.275
HbA1c (%)	6.65 ± 0.93	6.69 ± 0.68	6.62 ± 1.07	0.692
Patients with HbA1c ≤ 7.0% n (%)	74 (71.1)	33 (80.5)	41 (65.1)	0.090
AST (U/L)	20.19 ± 6.01	19.92 ± 6.77	20.37 ± 5.50	0.739
ALT (U/L)	22.18 ± 9.98	21.56 ± 9.80	22.59 ± 10.18	0.630
GGT (U/L)	29.38 ± 19.36	23.45 ± 10.68	34.38 ± 23.49	**0.040**
Total cholesterol (mg/dL)	163.49 ± 29.20	170.53 ± 29.97	159.11 ± 28.05	0.058
HDL-cholesterol (mg/dL)	49.32 ± 12.85	50.34 ± 18.23	45.52± 13.81	0.129
LDL-cholesterol (mg/dL)	88.30 ± 25.89	78.98 ± 38.29	84.56 ± 29.99	0.408
Triglycerides (mg/dL)	130.02 ± 60.45	131.15 ± 64.33	129.31 ± 58.41	0.882
Creatinine (mg/dL)	0.98 ±0.25	0.84 ± 0.18	1.01 ± 0.35	**0.007**
eGFR (mL/min/1.73 m^2^)	70.47 ± 23.04	70.05 ± 23.93	67.38 ± 26.78	0.606
Subjects with micro-vascular complications n (%)	28 (26.9)	11 (26.8)	17 (27.0)	0.999
Subjects with macro-vascular complications n (%)	53 (51.0)	21 (51.2)	32 (50.8)	0.966
Visits during 2-year period (n) *	3.61 ± 1.14	3.40 ± 1.20	3.75 ± 1.09	0.171
**Diabetes management**				
Hypoglycemic therapy changes (2) = years)	0.48 ± 0.69	0.47 ± 0.83	0.49 ± 0.60	0.852
Episodes of hypoglycemia n (%) **	7 (6.7)	4 (9.8)	3 (4.8)	0.321
Metformin n (%)	88 (84.6)	36 (87.8)	52 (82.5)	0.467
Acarbose n (%)	13 (12.5)	5 (12.2)	8 (12.7)	0.890
Secretagogues n (%)	31 (29.8)	15 (36.6)	16 (25.4)	0.220
DPP IV-i n (%)	24 (23.1)	12 (29.3)	12 (19.1)	0.227
Pioglitazone n (%)	3 (2.9)	0	3	-

Data are mean ± SD; n, %. eGFR: estimated glomerular filtration. * Number of specialist outpatient visits in the previous 2 years. ** Patients with at least one episode of hypoglycemia in the six months preceding the study. Significant *p* are reported in bold.

**Table 2 ijerph-19-13381-t002:** Tests scores at baseline in T2D elderly subjects participating to in the study, according to gender.

*Tests Scores at Baseline*	All	Women	Men	*p*
Mini Mental State Examination (MMSE)	25.61 ± 2.99	24.83 ± 2.98	26.11 ± 2.79	**0.028**
ADL (Activities of Daily Living)	5.80 ± 0.61	5.58 ± 0.76	5.95 ± 0.43	**0.002**
IADL (Instrumental Activities of Daily Living)	7.65 ± 0.91	7.53 ± 1.24	7. 75 ± 0.57	0.223
Diabetes Treatment Satisfaction Questionnaire (DTSQ)				
total score	30.84 ± 5.14	31.56 ± 5.25	30.33 ± 5.04	0.231
DTSQ 2	2.9 ± 1.80	3.26 ± 1.92	2.7 ± 1.70	0.125
DTSQ 3	1.6 ± 1.75	1.49 ± 1.59	1.69 ± 1.86	0.568
Diabetes Specific Quality of Life (DSQoL)				
*S* *atisfaction*	32.17 ± 9.35	35.24 ± 9.78	30.57 ± 8.46	**0.012**
*Impact*	36.89 ± 8.46	36.78 ± 8.86	36.97 ± 8.27	0.910
*Worry*	8.32 ± 2.89	8.76 ± 2.71	8.03 ± 2.98	0.210
Locus of Control (LOC)				
*Internal*	29.53 ± 6.33	29.49 ± 6.92	29.56 ± 5.97	0.958
*External*	24.63 ± 5.73	23.32 ± 6.19	25.49 ± 5.29	**0.048**
*Chance*	18.25 ± 7.70	18.83 ± 7.84	17.87 ± 7.65	0.539

Data are mean ± SD. DTSQ 2: perceived hyperglycemia; DTSQ 3: perceived hypoglycemia. Significant *p* are reported in bold.

**Table 3 ijerph-19-13381-t003:** Variation of clinical characteristics of T2D elderly subjects participating in the study, after 6 years of follow-up Metabolic control and hypoglycemic therapy in T2D elderly subjects participating to the study, after the 6 years-observation period.

	Baseline	Follow-Up	Delta (Δ)	*p*
Weight (kg)	73.73 ± 13.08	70.73 ± 13.77	−3.0 ± 0.066	**0.002**
BMI (kg/m^2^)	27.21 ± 6.42	26.79 ± 5.11	−0.42 ± 0.063	**0.003**
HbA1c	6.65 ± 0.93	6.66 ± 0.63	0.01 ± 0.10	0.561
Patients with HbA1c ≤ 7.0% n (%)	74 (71.1)	61 (58.6)	−0.17 (−17%)	0.492
eGFR (mL/min/1.73 m^2^)	70.47 ± 23.04	65.62 ± 21.17	−4.85 ± 0.29	0.451
Creatinine (mg/dL)	0.98 ± 0.25	1.14 ± 0.39	0.16 ± 0.31	**<0.001**
Subjects with micro-vascular complications n (%)	28 (26.9)	39 (37.5)	0.11 (+10.6%)	**<0.001**
Subjects with macro-vascular complications n (%)	53 (51.0)	69 (66.3)	0.16 (+15.4%)	**<0.001**
**Hypoglycemic therapy**				
Acarbose n (%)	13 (12.5)	10 (9.61)	−0.23 (−23%)	0.362
Metformin n (%)	88 (84.6)	60 (57.7)	−0.31 (−31%)	**<0.001**
Secretagogues n (%)	31 (29.8)	24 (23.08)	−0.22 (−22%)	**0.002**
DPP IV-i n (%)	24 (23.1)	38 (36.54)	0.58 (+58%)	**0.002**
Pioglitazone n (%)	3 (2.9)	2 (1.92)	−0.33 (33%)	0.313
GLP-1 RAs n (%)	0	6 (5.77)	-	
Insulin n (%)	0	6 (5.77)	-	
Intensification of diabetes therapy n (%) *	-	30 (28.8)	-	

Data are mean ± SD. Delta (Δ) values for continuous variables are expressed as mean and SD. Delta (Δ) values for categorical variables are expressed as variations in percentages. * Intensification of diabetes therapy: any increase in the number of OHA taken overtime by the patient or prescription of injection drugs (insulin or GLP-1 RAs). Significant *p* are reported in bold.

**Table 4 ijerph-19-13381-t004:** Clinical characteristics and tests scores at *baseline* in T2D elderly subjects participating to in the study, in the <Worsening control> and <Maintaining good control> groups.

Clinical Characteristics	All	Worsening Control Group	Maintaining Good Control Group	*p*
N (%)	104	64 (61.5)	40 (38.5)	-
Men	63	40 (62.5)	23 (57.5)	
Age (years)	71.88 ± 6.95	72.19 ± 6.92	71.40 ± 7.05	0.576
Diabetes duration (years)	12.50 ± 9.74	13.76 ± 11.19	10.55 ± 6.59	0.105
BMI (kg/m^2^)	27.21 ± 6.42	27.42 ± 4.35	28.25 ± 6.33	0.4354
Waist circumference (cm)	100.80 ± 11.39	99.23 ± 10.21	103.15 ± 12.87	0.2374
HbA1c (%)	6.65 ± 0.93	6.92 ± 0.61	6.39 ± 0.60	**<0.001**
Creatinine (mg/dL)	0.98 ± 0.25	0.97 ± 0.24	1.01 ± 0.27	0.443
eGFR (mL/min/1.73 m^2^)	70.47 ± 23.04	72.02 ± 24.36	68.00 ± 20.82	0.3964
** *Test scores* **				
Mini Mental State Examination (MMSE)	25.61 ± 2.99	25.75 ± 2.79	25.39 ± 3.11	0.534
ADL (Activities of Daily Living)	5.80 ± 0.61	5.98 ± 0.38	5.60 ± 0.87	**0.034**
IADL (Instrumental Activities of Daily Living)	7.65 ± 0.91	7.81 ± 0.50	7.43 ± 1.30	0.053
Diabetes Treatment Satisfaction Questionnaire (DTSQ)				
total score	30.84 ± 5.14	30.91 ± 5.25	30.73 ± 5.01	0.862
DTSQ 2	2.9 ± 1.80	3.02 ± 1.82	2.80 ± 1.79	0.5553
DTSQ 3	1.6 ± 1.75	1.55 ± 1.62	1.70 ± 1.95	0.666
Diabetes Specific Quality of Life (DSQoL)				
*S* *atisfaction*	32.41 ± 9.35	33.31 ± 9.42	30.35 ± 6.82	0.087
*Impact*	36.89 ± 8.46	37.67 ± 9.03	35.65 ± 7.42	0.238
*Worry*	8.32 ± 2.89	8.14 ± 3.05	8.60 ± 2.63	0.433
Locus of Control (LOC)				
*Internal*	29.53 ± 6.33	29.33 ± 5.94	29.85 ± 6.99	0.684
*External*	24.63 ± 5.73	**23.11 ± 4.79**	**26.40 ± 4.47**	**0.0007**
*Chance*	18.25 ± 7.70	18.09 ± 7.82	18.50 ± 7.61	0.795

Data are mean ± SD. <**Worsening control**> group (n = 64): Subjects who over time presented a worsening of glycemic control and /or requested an intensification of hypoglycemic therapy vs. baseline. <**Maintaining good control**> group (n = 40): subjects who over time improved their glycemic control or maintained a good glycemic control. DTSQ 2: perceived hyperglycemia; DTSQ 3: perceived hypoglycemia. Significant *p* are reported in bold.

## Data Availability

The data that support the findings of this study are available from the corresponding author, upon reasonable request.

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
