# Peer review of "Long-Term Influence of Locus of Control and Quality of Life on Metabolic Profile in Elderly Subjects with Type 2 Diabetes"

_ijerph, 2022, doi:10.3390/ijerph192013381_

Round 1

Reviewer 1 Report (Previous Reviewer 1)

Interesting work, it is no small task to track subject over 6 years. There are a couple of crossed out words and odd edits in the manuscript that need to  be cleaned up prior to publication. The interest with viewing the correlations would be to see where women vs men fell within the regression analysis, especially since none of the variables for metabolic control were associated the female gender. This an interesting difference that would be interesting to see if this hold in younger diabetic patients when estrogen is present. 

Author Response

We gratefully thank the Reviewer for the precious contribution. We hope to have significantly improved the overall quality of the manuscript, according to the Reviewer's suggestions. 

Reviewer 2 Report (Previous Reviewer 3)

Just some minor editorial input needed.

Author Response

We gratefully thank the Reviewer for the precious contribution. We hope to have significantly improved the overall quality of the manuscript, according to the Reviewer's suggestions. 

Reviewer 3 Report (New Reviewer)

Dear Authors,

The manuscript entitled “Long-term influence of Locus of Control and Quality of Life on metabolic profile in elderly subjects with type 2 diabetes” deals with an interesting and important topic, the manuscript is of high-quality. I have only a few minor recommendations to deal with.

When you are writing about differences observed between T2D men and women in lines 72-79, please, include the directions of differences, too (which gender showed higher/lower values). When you discuss various results of papers studying LOC (around lines 413-416), please, reveal some patterns (i.e., is there any connection between countries, age groups, etc., and if yes, what are the directions?).

The text is sometimes overly segmented, e.g., the two paragraphs in lines 67-76, as well as those in lines 80-85, 127-132, 151-156, and 403-408 should be merged. Waist circumference is unnecessarily repeated in Tables 1 and 4. Both tables as well as Table 3 contain some typos (e.g., p, kg, space before relational sign). At the bottom of Table 2, some categories are not centered. The title of Table 3 is not appropriate. When including p values, please use the same number of decimal places everywhere (see, e.g., Table 4, which differs from the other tables in this respect). Since LOC-E was not defined previously, please, use the term “LOC-External” in line 335. You should use the term “because of” or “due to” but not both of them in line 493. The term “American” is repeated in line 609.

There are some typos in the text, see, e.g., lines 44, 52, 70, 73, 78, 93, 109, 111, 130, 203, 232, 238, 260, 261, 266, 269, 271, 274, 277, 278, 279, 280, 283, 294, 307, 329, 332, 336, 347, 349, 352, 353, 354, 355, 358, 359, 364, 386, 393, 405, 423, 428, 430, 487, 507, 522, 629, 631, and 666. There are some formatting issues as well, e.g., in lines 9 and 409 the use of bold letters is unnecessary; indent first line is missing in line 330.

Author Response

We gratefully thank the Reviewer for the precious contribution. We hope to have significantly improved the overall quality of the manuscript, according to the Reviewer's suggestions. 

This manuscript is a resubmission of an earlier submission. The following is a list of the peer review reports and author responses from that submission.

Round 1

Reviewer 1 Report

The purpose of the current manuscript by Giandalia and colleagues was to determine if the QoL and LOC have a predictive role in determining metabolic control and diabetes outcomes in elderly individuals with type 2 diabetes. While the study potentially has some usefully information for the clinic where the observations took place, the generalization of the study results is questionable. This is due mainly to the limited information concerning the study population. The authors hinted at some of the study population details in the discussion, but this needs to be addressed in the methods and results sections. The study examined >65 yr old individuals who received T2DM care at one facility in Italy but other that very little information is given about the sample population. It is unclear in the study whether the study sample (N=104) is sufficient to represent the target population/application of a predictive model. A suggestion would be for the authors to provide a power analysis. Additionally, subject qualifiers for example: urban/rural dwellers (live within XX miles of facility), live alone/co-habitat/care facility, married/single, income, ethnicity, et cetera would be helpful for the interpretation of the results especially given the fact that the authors observed some difference in LOC-E.   

The secondary goal of the study was to examine if sex alters QoL and LOC in the study population.  This would potentially increase the novelty of the study and is needed in the literature. However, as the authors point out the study has a small sample size for determine sex difference thus it limits the studies ability to make any solid conclusions in this regard. While a suggestion would be to increase the sample size to make a conclusion based on sex, it is understood that a 6-year duration of a study is not a small undertaking. Thus, given the limited N for determining sex difference and the fact that it is a secondary aim of the study change the manuscript title and conclusion to represent the strongest conclusion of the study which is in elderly T2D patients in Italy.

Minor comments:

- Subjects: >65 years at metabolic outpatient clinic.

- Line 81: Suggests that “validated questionnaires” were utilized, however there are several validated questionnaires that measure ADL for example. In the text, please indicate which questionnaire was used and reference the validation study.

- Indicate how the questionnaires were given (written, oral if so by who, ect).

- Please provide scoring meaning for the ADL and IADL. What is a max score?

- IADL previously has shown to have a gender bias, due to gender expectation within a culture. This is something that should be addressed with the currently study population and indicated how this may impact interpretation of results.

- In the method section “2.5 Assessment of T2DM chronic complications” it is unclear how “micro- and macrovascular complications” were applied to the study. Were both micro- and macro- complications considered “new” occurrence of a complication or was just macro- considered. Please expand on explanation of how results were obtained (Line 224).

- Table 1: Please double check “Diabetes duration” years in the “Men” group.

- Show logistical regression graphs for regression analysis.

- Line 257: if you have obtained that data and are going to refer to the results, please put the data in the manuscript. “Data not shown,” isn’t appropriate.

- Line 220 and Line 239. Line 220 indicates that only 30 subjects following 6-years required increase in treatment yet the way line 239 read it seems like the worsening control group had an N=64. Please provide more detail how the worsening control group was defined. Additionally, it would be helpful if this information was in the Figure 1 legend.

- Throughout the discussion remind reader of the population, 1 clinic in Italy.

Reviewer 2 Report

Thank you for the invitation to review this article. Please, allow me to share some comments and suggestions.

Abstract

I am not so familiar with some definitions; therefore, it would be interesting if the authors could clarify some terms, like “locus of control” (internal and external?) and define abbreviations, like DSQoL

Introduction

The paragraphs are too short. I suggest the authors to merge some of them to improve readability.

Please define QoL before presenting the abbreviation.

The introduction needs more details about the hypothesis and the rationale. What is the rationale to evaluate the association between QoL and LOC on metabolic control (it seems more plausible that metabolic control influences QoL and not the opposite)? Why studying older people? You compare men and women but did not provide rationale for the that. Is there any gap that you would like to fill?  Many questions remain without answer after reading the introduction.

Results

Lots of focus in comparison between sexes, but there is no previous presentation of this analyzes. Please provide a rationale for comparing men and women in the introduction.

Don’t the authors think counterintuitive that people with higher scores o LOC external showed better metabolic control? And how about the association with ADL and metabolic control?

Discussion

Lines 329-337 – lots of studies have made similar analysis (there is even a meta-analysis about the topic), what is the difference from them and the present one? Why this study would worth being published? Longer follow-up? Better control? More details? Please provide this information in the introduction and discussion. It is also important to discuss why the present study has different results from the others.

Lines 404-413 – lack of LOC follow-up is an important limitation. Maybe this changed after the patient receives specific counselling. Another important limitation is the lack of control for behavioral changes, since diet and physical activity are two important factors for glucose control.

Reviewer 3 Report

The study evaluated the predictive role of LOC and QoL scores on metabolic control in elderly T2D outpatients, evaluating potential gender differences. While this is an interesting approach, authors need to address the following issues:

1. The paper requires extensive editorial input. For example, Line 23 (need a punctuation mark after 'analysis'), Line 92 (needs revision), Line 96 ('toileting' can be reworded), Line 98 (making a shop can be reworded), Line 98 needs 'and' before 'cleaning houses', Line 108 ('into three'), Line 169 (revise) etc

2. Be consistent with the use of T2D or T2DM and HbA1c/Hba1c etc.

3. Line 181: Total cholesterol tended to be different

4. Line 191: Double check the numbers on the table as Secretagogues was taken by 9 females plus 16 males, amounting to 25 and not 31. 

5. Double check all the numbers on the Tables (e.g., Diabetes Duration on males in Table 1 does not look right).

6. Effect of the different Treatment on glycaemic control needs to be highlighted as well.

7. Line 297: HbA1c is a measure of glycaemic control. My understanding is that the psychological aspects contribute to treatment outcome, thus, rather, replace 'HbA1c targets' with 'treatment measures'.